# Impact of the Rise of Artificial Intelligence in Radiology: What Do Students Think?

**DOI:** 10.3390/ijerph20021589

**Published:** 2023-01-16

**Authors:** Andrés Barreiro-Ares, Annia Morales-Santiago, Francisco Sendra-Portero, Miguel Souto-Bayarri

**Affiliations:** 1Department of Radiology, School of Medicine, University of Santiago de Compostela/CHUS/IDIS (Instituto de Investigación Sanitaria de Santiago), 15782 Santiago de Compostela, Spain; 2Department of Radiology and Physical Medicine, School of Medicine, University of Malaga, 29010 Málaga, Spain

**Keywords:** medical students, radiology, artificial intelligence

## Abstract

The rise of artificial intelligence (AI) in medicine, and particularly in radiology, is becoming increasingly prominent. Its impact will transform the way the specialty is practiced and the current and future education model. The aim of this study is to analyze the perception that undergraduate medical students have about the current situation of AI in medicine, especially in radiology. A survey with 17 items was distributed to medical students between 3 January to 31 March 2022. Two hundred and eighty-one students correctly responded the questionnaire; 79.3% of them claimed that they knew what AI is. However, their objective knowledge about AI was low but acceptable. Only 24.9% would choose radiology as a specialty, and only 40% of them as one of their first three options. The applications of this technology were valued positively by most students, who give it an important Support Role, without fear that the radiologist will be replaced by AI (79.7%). The majority (95.7%) agreed with the need to implement well-established ethical principles in AI, and 80% valued academic training in AI positively. Surveyed medical students have a basic understanding of AI and perceive it as a useful tool that will transform radiology.

## 1. Introduction

The rise of artificial intelligence (AI) in all fields of medicine and, particularly, in radiology, is gaining more and more importance. Its scope will transform the way of medical practice, as well as the education model for the future. In many cases, the performance of health professionals will depend on the proper knowledge and use of AI, which is expected to lead to the provision of quality health care, based on precision medicine, comprehensive diagnosis and personalized prognosis for the benefit of the patient.

The concept of AI has been in continuous evolution because of the development and interactions between the different systems that compose it. In the context of medical activity, AI means the development and use of algorithms and software techniques that can interpret medical data, in order to learn from them, adapt and improve their performance through this experience, and thus helping increase patient safety and reducing the direct workload of professionals [1,2,3,4,5,6].

Its applications in radiology are multiple and can range from scheduling appointments to patients [7] and selecting imaging protocols for optimal radiation doses [8], to other different tasks that have to do with image processing (reconstructions, quality improvements, lesion detection, measurements, organ segmentation, documentation) [9,10].

The COVID-19 pandemic has given a boost to the research sector, favoring the integration of this technology in digital radiology [11]. Its applications in computed tomography/X-ray of the thorax seem to bring the moment of its introduction in the clinical routine [12,13]. New hybrid networks have been investigated, a combination of machine learning (ML) and DL (deep learning), which will allow prediction of the survival or severity of the diseases by integrating the characteristics of the lesions with the clinical and laboratory data of the patients, a discipline which is still under investigation [9,14,15].

The use of radiomic models, which use advanced ML approaches and imaging characteristics, for example, magnetic resonance imaging, are methods that are beginning to be spread, because they are simple and non-invasive, and may help in diagnosis and therapeutic decision-making, and in personalizing them, such as in prostate cancer [16]. Other models correlate different imaging studies (CT, PET) with data (proteomics, genomics, liquid biopsy, etc.) that help predict prognosis and response to treatment [2,17].

Despite the efficacy demonstrated by AI in some specific tasks, the radiologist’s work is far from being replaced globally [18]. It is believed that the model that will prevail will be the use of this technology as support [19,20], for which communication and joint work with other professionals such as engineers and computer scientists [21], and their adequate training, will be crucial. AI will have a positive impact on the work of radiologists, since they will not only be seen as image interpreters, but will also lead the algorithm validation process and contribute their experience in the global clinical approach to patients, obtaining greater relevance and visibility.

Few studies have been carried out in recent years addressing the need for AI education for both students and residents, as well as medical specialists. There seems to be a generalized and growing consensus to continue training radiologists who incorporate new knowledge and skills in said technologies, and that this training should begin in the university phase, be consolidated in residency and continue as part of continuous training throughout the profession [19,22,23].

There are many training programs offered by a wide range of institutions, but they are often occasional, short, and not integrated into the radiologists’ learning path. The fact that AI training is only recently emerging creates a significant gap between what these programs offer and what radiologists need to learn [24,25]. What is clear is the need for adequate training that includes the use, benefits, challenges and issues related to the implementation of AI in clinical departments to ensure that it increases the confidence of clinicians interested in these careers [26]. Students should be engaged in practical exercises with real applications of AI and learn to use it effectively and critically in their work, in the same way that younger radiologists should be supported by programs that allow them to strategically design and develop their professional career for the future [24].

Thus, the principal aim of this study is to analyze medical students’ perception about the current situation of AI in medicine, especially in radiology. Secondly, and in a complementary way, other objectives would be:To check the general knowledge that students have about AI.To assess the importance given by students to academic training in AI.To determine the influence of AI on human decision-making and capabilities, as well as the need for the implementation of well-established ethical principles.To investigate the role assigned to the use of AI in radiology and its impact on the professional performance of specialists and radiology services.

## 2. Materials and Methods

To carry out this work, the process was divided into two parts:Theoretical: A review of the subject to be investigated was accomplished through different search engines such as: PubMed, Scopus, Dialnet and Google Scholar. Articles from the last 5 years were included, limiting the references to our objectives using the keywords: artificial intelligence; medicine; radiology; students; perception; ethics.Empirical: The tool used was an anonymous online questionnaire (Appendix A), carried out through the Google Forms platform to high school and university Medical students in Santiago de Compostela during the 2021–2022 academic year. The distribution of the survey was made through the class delegates in the Faculty of Medicine and through the directors of the institutes. It was open from 3 January to 31 March 2022.

The questionnaire consisted of 17 items including multiple choice questions, true/false questions, and 5-point Likert scale items. Questions tried to address the following main topics: (1) Demographics (sex, age, current course, and institution) (2) Ranking radiology specialty with or without the consideration of AI (3) Assessment of their confidence in and understanding of AI (4) Sources of information on AI (5) AI and ethics (6) Asses their opinion about teaching the basis of AI in universities (7) Perception of the impact of AI in radiology.

The questionnaire was conducted as an empirical basis for two different research projects. One sought to analyze the perception of AI in high school students, and another in medical students.

For this reason, the total sample size was 402 respondents, of which the data of the high school students (*n* = 119) were not included in this study, as our objective was to seek medical students’ perceptions. Thus, the statistical analysis was performed on the responses of the University Degree in Medicine group (*N* = 283).

Participation was voluntary and had no relation to the student’s curricular activities. Informed consent was confirmed when a respondent agreed to start the online survey. Each participant was informed that their data would be anonymous, statistically analyzed and used for scientific publication.

Data analysis was performed with the statistical analysis software R for Mac (4.1.2) with R Studio. First, a descriptive analysis of the data was carried out using absolute and relative frequencies.

Likert responses were converted to ordinal data by assigning the following values: strongly disagree = 1; disagree = 2; neither agree/nor disagree = 3; agree = 4; and strongly agree = 5. They were treated as ordinal data, using percentages to analyze them, and using the ggplot2 and likert packages of R to plot them.

Age did not follow a normal distribution.

Next, it was verified how the variables age and sex influenced the results (the chi-square test and the Mann-Whitney-Wilcoxon test were performed as appropriate).

Likewise, comparisons were made between some of the rest of the variables.

A *p*-value <0.05 was regarded as statistically significant.

## 3. Results

### 3.1. Demographics

This work is based on a totally anonymous online questionnaire, given to students studying a medical degree during the 2021–2022 academic year. The information set out below has been obtained from the analysis of their answers.

A total of 283 responses were obtained, of which 281 were valid, and two were invalid due to incorrect filling out.

According to age (22.2 ± 3.5), the sample was divided into three age groups from highest to lowest frequency:-Group 1. Between 21 and 22 years old: With 42.34% (*n* = 119).-Group 2. Age equal to or less than 20 years: Represented by 34.16% (*n* = 96).-Group 3. Age equal to or greater than 23 years: 23.48% (*n* = 66) of the students belong to this group.

Regarding gender, 71.17% (*n* = 200) were women, the majority, compared with 28.83% (*n* = 81) of men (Table 1).

The distribution of students in different courses, from highest to lowest participation, showed that 25.62% (*n* = 72) corresponded to the 6th grade, followed by the 2nd and 5th grades with the same number of students, respectively, 50 (17.80%), 13.52% from 1st year (*n* = 38), 12.81% (*n* = 36) correspond to those from 3rd and, finally, those from 4th that represent 12.45% (*n* = 35) of the sample.According to the origin of the respondents, the majority belonged to the University of Santiago de Compostela (USC), making up 97.86% (*n* = 275). The rest came from the University of Murcia, with 1.06% (*n* = 3), the University de las Palmas de Gran Canaria (ULPGC), with 0.36% (*n* = 1), UCS, with 0.36% (*n* = 1) and the UNED, with another 0.36% (*n* = 1).

### 3.2. Ranking Radiology

When asked about the choice of radiology as a specialty, 75.09% (*n* = 211) of those surveyed answered that they would NOT choose it, compared with 24.91% (*n* = 70), that YES, they would choose it. Age was associated with a greater probability when choosing radiology as a specialty *p* = 0.007 (*p* < 0.05). These results were independent of gender and the course in which they were (*p* = 0.31 and *p* = 0.13, respectively).Among the students who wanted to do radiology, 6.74% (*n* = 6) said that they would choose radiology as the first option, 3.37% (*n* = 3) as the second option, 30.34% (*n* = 27) as the third option and 59.55% (*n* = 53) said they would be indifferent to the order of preference.At the end of the questionnaire, we asked them if they would change their choice considering the impact of AI, and we found that 159 (56.58%) would not change their preferences when choosing specialty, regardless of the impact that AI would have on them, while only 27 (9.61%) would and 95 (the remaining 33.81%) said that perhaps they would reconsider their choice.

No statistically significant differences were found regarding the gender variable (*p* = 0.26).

Regarding question No. 1 (Choice of Radiology) and the change of preference after thinking of the impact of AI, a statistically significant difference (*p* = 0.0039) was observed. It seems that those who chose radiology were more aware of the potential impact of AI on radiology and were more hesitant.

### 3.3. Knowledge of AI

To assess the students’ knowledge of AI, questions were asked that assessed the subjective perception of knowledge, and then exposed them to statements of true and false to verify their knowledge in a more objective way.

#### 3.3.1. Subjective

They were first asked if they knew what AI was and its uses.

79.36% (*n* = 223) of the respondents answered affirmatively, 10.32% (*n* = 29) said NO and the other 10.32% (*n* = 29) chose “I don’t know/I don’t answer (NS/NA)”, Table 2.

A statistically significant result (*p* = 0.009) was obtained regarding the gender variable. It seems that men are more confident when answering YES, compared with women, who usually answer NS/NA.

No statistical difference was found between question No. 1 (Choice of Radiology) versus No. 3, (*p* = 0.72).

Regarding the use of AI in our daily lives (voice and face recognition systems, web search engines, cybersecurity, autonomous vehicles, robots, online shopping, advertising…), 81.14% (*n* = 228) answered YES, 16.72% (*n* = 47) chose NO as an answer, and the remaining 2.14% (*n* = 6) chose NS/NA in terms of being aware of the use of AI in daily life.As for the source from which they obtained information, MEDIA was the most chosen (250; 88.97%), followed by articles and journals (118; 41.99%), friends/family (104; 37.01%), teachers/center (101; 35.94%), radiologists (29; 10.32%) and other specialists (21; 7.47%).

This question accepted that respondents (*n* = 281) checked more than one option. Analyzing the data collected about the sources used by students to obtain information about AI, we proceeded to organize the options from highest to lowest relevance (number of times marked) for better understanding.

#### 3.3.2. Objective

Then we asked them true/false questions to assess their objective knowledge.

As we see in Table 2 and Figure 1, success rate in true questions was 65,1%. Additionally, the percentage on false questions was low.

### 3.4. General Perception of the Impact of AI

We asked two general questions about the impact of AI on human activities.

A total of 56.23% (*n* = 158) stated that AI improved human capabilities. Meanwhile, 38.08% (*n* = 107) stated that it increased them and only 5.69% (*n* = 16) answered that it did not influence these capacities.When asked whether AI could affect human autonomy by interfering with decision-making, 46.99% (*n* = 132) agreed and 8.13% (*n* = 23) strongly agreed.

25.09% (*n* = 70) were neutral with this issue (Neither agree/Nor disagree), reflecting insecurity in answering this question.

The rest of the participants stated that it would not affect decision-making, 15.90% (*n* = 45) disagreed and 3.89% (*n* = 11) strongly disagreed.

### 3.5. Perception of the Impact of AI in Radiology


91% of respondents said AI will change the way radiologists work [Agree, 70.46% (*n* = 198) and Strongly Agree, 20.28% (*n* = 57)]. Figure 2.Most of the respondents, 90.39% (*n* = 254), assigned a support role to the use of AI in radiology, followed by 8.90% (*n* = 25), who opted for preponderant and only 0.71% (*n* = 2) were against its use.99.64% (*n* = 280) agreed that the main role of radiologists should be “Lead the algorithm validation process, contribute their experience in the global clinical approach of patients, and make the final decision”.In question 13 we explored the fear of replacement.


A significant number of respondents expressed themselves against AI being able to replace radiologists, where 46.62% (*n* = 131) disagreed and 33.10% (*n* = 93) strongly disagreed.

A minority chose Strongly Agree, 2.49% (*n* = 7), or Agree, 8.54% (*n* = 24).

The remaining 9.25% (*n* = 26) did not position themselves, and they marked Neither agree/Nor disagree.

Finally, we investigated what aspects of the radiology service would improve with AI.

As this question was multiple choice for its analysis, we organized it from most- to least-chosen by the respondents:It favors early diagnosis and treatment of diseases: 244 (86.83%).It improves the management and quality of radiology services: 193 (68.68%).It allows radiologists to focus on patient care in a general clinical context by making their work easier: 137 (48.75%).It reduces the number and qualification of the professionals needed in the service: 27 (9.61%).

### 3.6. Ethics

Most students agreed that AI should follow ethical principles. 74.73% (*n* = 210) said they totally agree and 21.00% (*n* = 59) agree. Figure 2.

### 3.7. Teaching

When asked if they believed that students should be trained in the use of AI, most respondents were in favor (80%) (*n* = 225), with 36.65% (*n* = 103) Strongly Agreeing and agreement 43.42% (*n* = 122).

Neither agree/Nor disagree (Neutral opinion), had 13.52% (*n* = 38).

A total of 18 respondents, the minority, took a stand against this formation. Disagree 1.78% (*n* = 5) and totally disagree 4.63% (*n* = 13).

### 3.8. Drawbacks of Using AI in Medicine

In the analysis of this multiple-choice question regarding the biggest drawback of the use of AI in medicine, the answers marked from the majority to minority opinions were organized:

They cannot interpret the patient in a global clinical context: 221 (78.65%).

High cost of its implementation: 116 (41.28%).

Possible vulnerability of the right to privacy of patients: 112 (39.86%).

Necessary training in the management of AI for professionals: 107 (38.08%).

No further statistically significant differences were found across the variables.

## 4. Discussion

Most of the students surveyed knew what AI was and its uses, as well as being aware of the applications of this technology in daily life. These data are part of their self-perception on the subject, being objectively demonstrated through True/False questions on basic knowledge, with an acceptable percentage of correct answers.

A previous study, carried out with Spanish medical students in 2021, yielded very similar results, both in self-perception and in the objective evaluation of knowledge, where 93% reported knowing the applications of AI in daily life and the correct rate of questions on basic knowledge was 70.7% [27]. Nearly half of UK students objectively demonstrated their basic knowledge [28]. In a Canadian study, a large number of the respondents reflected a high level of confidence in their understanding of AI; in contrast, only half correctly answered at least 3 of the 5 questions that were asked to objectively determine their knowledge [29].

At this point, we find studies, such as those from Saudi Arabia, where knowledge about AI is low, and 54.4% of the students were not able to correctly answer any of the basic questions, despite self-perceiving knowledge of the subject [30]. Almost 63.1% of the Germans surveyed considered themselves tech savvy, but only 1/3 claimed to have basic knowledge of AI (30.8%) [31].

The majority of respondents (95.73%) in this study were in favor of the need for the implementation of well-established ethical principles in AI. This point has not been explored in the studies analyzed, despite the repercussion that the impact of AI has. It would be convenient that, with a view to the future, questions of this type were incorporated, in order to compare the perception between different population groups in this regard.

A significant number of students (80%) positively valued academic training in AI. Most of the studies reviewed agree on this point, as respondents believe that teaching AI will be beneficial for their career and agree to include it as part of their study program [27,28,29,30,31,32,33,34]. The fact of not receiving education in the use of new technologies could affect students, when choosing radiology or not, feeling prepared to work with them at the end of their degree [28].

The applications of this technology are valued positively by the majority of students, who give it an important support role. Approximately half of the students surveyed see AI as a useful tool that improves human capabilities, despite affecting our autonomy by interfering with decision-making.

According to a UK study, 88% of its students believed that it will play an important role in the medical care of the future and 48.3% believed that certain specialties will be replaced (36), while others believed that AI will revolutionize radiology, as in the case of the German article (77.2%) or ours (90.74%) [27,31]. For 66% of American students, AI will have a profound impact on medicine in general and especially radiology [32].

In turn, 87.7% of the students surveyed by Galán and Portero in 2021 said that “advances in AI will improve the workload capacity and efficiency of radiologists, constituting a very useful complement to their work” [27].

A large number of students (79.72%) considered that the rise of AI could not replace radiologists. The vast majority of those surveyed felt that the way radiologists work would change. Respondents from the German and Spanish studies, for the most part, stated that it will not replace doctors but that in the case of radiology it could be seen as a good competitor [27,31]. Just over half of Saudi and Canadian students did not agree with AI replacing radiologists, and 44.8% and 67.7%, respectively, believe that in the future the number of these specialists will be reduced by AI [29,30]. A total of 90.2% of Canadian students stated that radiologists should adopt AI and work with the IT industry for its application; on the other hand, they agreed with the view of “displacement”, due to greater efficiency with the increase of the AI [29].

As the greatest drawback of the use of AI in medicine, the respondents pointed out the inability to interpret the patient in a global clinical context, followed by the high cost of its implementation, the possible vulnerability of the right to privacy of patients and the necessary training in the management of AI for professionals.

All the studies believe that it would be necessary to have training in AI both at the student and professional level. German respondents stated that AI could potentially detect pathologies in radiological examinations (83%), but it could not establish a definitive diagnosis (56%) [31], for which the role of the radiologist is vital.

Although the results obtained in the research were fruitful, it is necessary to mention a limitation was found as far as the study sample is concerned: the size is relatively small if we take into account the number of medical students in Spain, and the origin, almost in its entirety, is from one university (USC). It is recommended to carry out new studies with samples from other universities, which include other questions that allow a better assessment of the perception that students have on the subject. In addition, emphasizing the need to train students in AI, including it as a subject in the degree’s study plan, as well as encouraging the choice of specialty in classes and providing it with a greater presence in clinical practice rotations will also be important.

## 5. Conclusions

This paper analyzed the perception that medical students had about the current situation of AI in this field and especially in radiology. With the data obtained and their analysis, it was possible to conclude:Applications of AI in medicine, and especially in radiology, are positively valued by the vast majority, considering it a useful tool.A high percentage of students have an acceptable knowledge of what AI is and its applications in daily life, however it could be improved. MEDIA was the main source of information.Most of the students consider their academic training in this discipline of vital importance for the future.They also agree on the need to implement well-established ethical principles in the field of AI.Most of them agree that the impact of AI in the specialty will not replace radiologists, but their work will undergo modifications.

Finally, they raise the impossibility of interpreting the patient in a global clinical context as the main drawback of AI in medicine.

## Figures and Tables

**Figure 1 ijerph-20-01589-f001:**
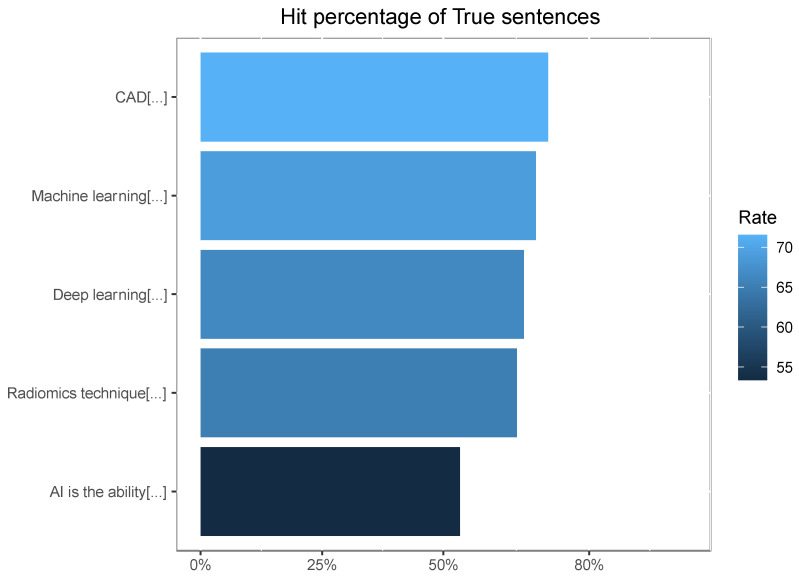
Percentage of correct answers in true sentences with an average accuracy of 65.1%. Abbreviations of True sentences used in Figure 1 due to their length: CAD [(computer-aided diagnosis): these are computer-aided diagnosis tools developed to detect, to segment and to classify lesions or complex patterns in radiological images]. Machine learning [(automatic learning) allows machines through algorithms and mathematical models to learn without being expressly programmed for it]. Deep learning [techniques based on artificial neural networks that process data and are capable of automatically recognizing patterns in biomedical images]. Radiomics technique [that consists of obtaining quantifiable information from medical images such as magnetic resonance, computed tomography or PET. They are important in detecting, evaluating and monitoring diseases]. AI is the ability [of advanced computer systems to perform the same tasks as human beings (with capabilities such as: reasoning, learning, creating and planning)].

**Figure 2 ijerph-20-01589-f002:**
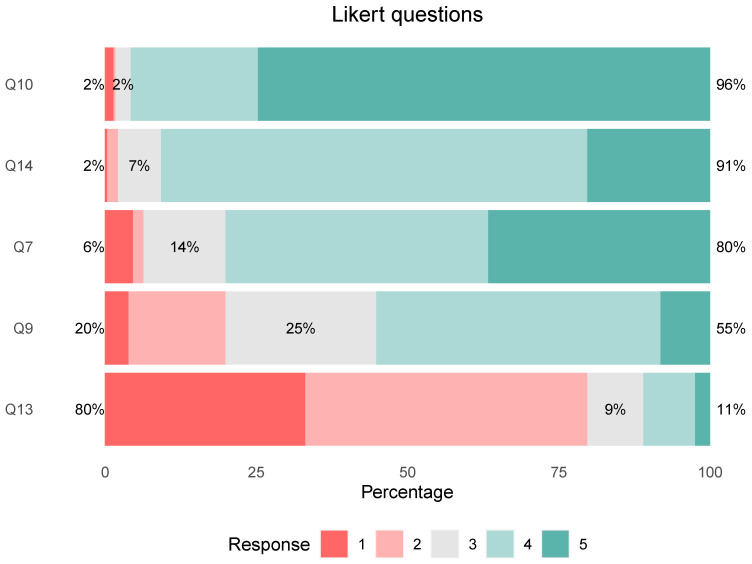
Analysis of Likert questions with horizontal bar chart. A high percentage of agreement (Agree and Totally agree) in ethics, education and impact of AI in radiology was observed. A high percentage of disagreement 80% (Disagree and Strongly disagree) was about the fact of radiologists being replaced by AI. Abbreviations used in Figure 2 due to their length: Q10 (Do you think AI should follow ethical principles?); Q14 (Will the way radiologists work change with the impact of AI?; Q7 (Do you think that students should be trained in the use of AI?); Q9 (Do you think AI can affect human autonomy by interfering with decision-making?); Q13 (Do you think the rise of AI could replace radiologists?). Response: 1 = Strongly disagree; 2 = Disagree; 3 = Neither agree/Nor disagree; 4 = Agree; 5 = Totally Agree.

**Table 1 ijerph-20-01589-t001:** Demographics.

Demographics		*N* = 281
Sex		
	Men	81 (29%)
	Women	200 (71%)
Grade		
	1	38 (14%)
	2	50 (18%)
	3	36 (13%)
	4	35 (12%)
	5	50 (18%)
	6	72 (26%)
Age		22.2 (3.5) ^1^

^1^ Mean (SD).

**Table 2 ijerph-20-01589-t002:** Subjective and objective assessment of the students’ AI knowledge.

Objective Knowledge Based on TRUE/FALSE Statements		*N* = 281	%
Radiomics emerged from the fields of radiology and oncology and its application is exclusive to them. A(answer)/False.		261	93
The use of deep learning in radiology does not require large databases of medical images for good pattern recognition. A/False.		261	93
CAD (computer-aided diagnosis): these are computer-aided diagnosis tools developed to detect, to segment and to classify lesions or complex patterns in radiological images. A/True.		201	72
Machine learning (automatic learning) allows machines, through algorithms and mathematical models, to learn without being expressly programmed for it. R/True.		194	69
Deep learning: techniques based on artificial neural networks that process data and are capable of automatically recognizing patterns in biomedical images.A/True.		187	67
Radiomics: technique that consists of obtaining quantifiable information from medical images such as magnetic resonance, computed tomography or PET.They are important in detecting, evaluating and monitoring diseases. A/True.		183	65
AI is the ability of advanced computer systems to perform the same tasks as human beings (capabilities such as: reasoning, learning, creating and planning). A/True.		150	53
**Subjective knowledge:** Do you know what artificial intelligence (AI) is and its applications? (Q3)	**Men**, *N* = 81	**Women**, *N* = 200	***p*-value**
No	6 (7.4%)	23 (12%)	0.009
I don’t know	2 (2.5%)	27 (14%)	
Yes	73 (90%)	150 (75%)	

## Data Availability

Not applicable.

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
