# Peer review of "Impact of the Rise of Artificial Intelligence in Radiology: What Do Students Think?"

_ijerph, 2023, doi:10.3390/ijerph20021589_

Round 1
Reviewer 1 Report
The manuscript (ijerph-2144867) titled “Impact of the rise of Artificial Intelligence in Radiology: What do students think?” analyzes the perception of medical students about the growing interest in Artificial Intelligence applications in Medicine and especially in the medical specialties of Radiology. To do this, it uses a survey made up of 17 items carried out on medical students distributed among the different courses of the Degree. Currently, the use of artificial intelligence in medical applications is of significant interest to health professionals.
The submitted manuscript is original and simple. The authors carry out an adequate bibliographic search, a correct statistical analysis, a clear exposition of the results, an updated discussion and some conclusions that emanate from the results obtained.
Minor considerations
In my opinion, the survey with its 17 items that has been incorporated into the manuscript as Appendix A could be incorporated as an independent Supplementary Materials file following the characteristics of [IJERPH].
In my opinion, it is an acceptable manuscript for publication.
Author Response
Thank you very much for your commentaries and suggestions. As you suggest we will include the survey as a Supplementary Material file (we have changed it in the text). We will upload the file as soon as possible.
Reviewer 2 Report
- It is necessary to explain why only students from one institution have participated for the most part. There is a testimonial participation from other institutions.
- It is interesting to know the participations by grade (Table 1), but it would be interesting to indicate if the authors are teachers in any course and if this has influenced participation.
- Explain clearly the reason for discarding the responses from the high school and what type of high school was invited to participate.
- In Figure 1 it is clearer to indicate to which question number each answer corresponds, as indicated in Figure 2 (Q1..Q3). The same for Table 2 and the detail of results (Section 3.4). The questions referred to (Q1..Qn) should be indicated when describing the results.
- In Figure 3 the color of the answers of value 3 should be changed because they are confused with the color of the background of the graph.
- The question in section 3.6 should be reflected upon. It is deduced that there are students who believe that IA should be applied without ethical principles.
Bibliographic references: should be revised
- Bibliographic reference 10 seems incorrect. It is a 2018 article and the DOI refers to a figure. I understand that it refers to the full article doi:10.1007/S11604-018-0796-2 .
- The DOI of reference 2 is incomplete: 10.1186/s13244-019-0738-2.
- The DOI of reference 4 is doi:10.1186/S13000-021-01085-4
- The DOI of number 16 is 10.1007/s11547-018-0966-4
- Review issues 17, 24, 26, 26, 28, 31
Suggestions
Seek mechanisms to achieve a greater representation of participants from different institutions.
Use more secure platforms such as Redcap (https://www.project-redcap.org/) as a tool for distributing surveys.
In relation to interest in the specialty of radiology, the relationship between students who have taken the specialty subjects and those who have not should be studied.
Author Response
Dear reviewer,
Thank you very much for your comments and suggestions. Please see the attachment to view our answers.

Reviewer 3 Report
The manuscript entitled " Impact of the rise of Artificial Intelligence in Radiology: What do students think?" debates the opinions of a group of students regarding the use of Artificial Intelligence in Radiology. While the study is interesting and the approach is valid, some aspects should be addressed:
1. The Abstract should be designed according to the journal's demands and template.
2. The aims of the study are well defined in the Introduction section. However, the Methods section and the Results section do seem to entirely follow all those aims mentioned here.
3. In addition, as one of the part in the Methods section is to "1. Theoretical: A review of the subject to be investigated was accomplished through different search engines such as: PubMed, Scopus, Dialnet and Google Scholar. Articles from the last 5 years were included, limiting the references to our objectives using the
keywords: Artificial Intelligence; Medicine; Radiology; Students; Perception; Ethics." I would expect to see in the Results section at least the results of the literature review, and what is that the literature is mentioning regarding this issue.
4. Also, in Methods section, I did not find any information about how was the questionnaire validated before being used as a tool in this study; in addition, the paragraph "The total sample size was 402 respondents, of which the data of the high school students (n=119) were not included in this study and the statistical analysis was performed from the responses of the University Degree in Medicine group (n = 283)" is confusing. More information should be provided about the group sample.
5. The sentence "A total of 283 responses were obtained, of which 281 were valid and 2 invalid due to incorrect fill out" should not be included in the Methods section, as it is already one of the Results outcomes. Also, the following lines 123-130 should be reconsidered as there seems to be part of the Results Section. In the Methods section, I would advise introducing a reference for "Likert questions" and properly explaining the Data analysis process.
6. In the Results section, more inferential statistics should be introduced, as the process mentioned in the Methods section.
7. The Conclusion section should be designed according to the aims of this study and to answer all the objectives mentioned there, in the Introduction section.
Author Response
Dear Reviewer,
Thank you very much for your comments and suggestions. Please see the attachment to view our answers.

Round 2
Reviewer 3 Report
The manuscript has been improved